# Network Pharmacology and Experimental Validation Identify Paeoniflorin as a Novel SRC-Targeted Therapy for Castration-Resistant Prostate Cancer

**DOI:** 10.3390/ph18081241

**Published:** 2025-08-21

**Authors:** Meng-Yao Xu, Jun-Biao Zhang, Yu-Zheng Peng, Mei-Cheng Liu, Si-Yang Ma, Ye Zhou, Zhi-Hua Wang, Sheng Ma

**Affiliations:** 1Department of Nuclear Medicine, Tongji Hospital, Tongji Medical College, Huazhong University of Science and Technology, Wuhan 430030, China; xumy1129@163.com; 2Department of Urology, Tongji Hospital, Tongji Medical College, Huazhong University of Science and Technology, Wuhan 430030, China; m202376512@hust.edu.cn (J.-B.Z.); hhpyzyyds@163.com (Y.-Z.P.); meicheng_liu@163.com (M.-C.L.); tjmasiyang@163.com (S.-Y.M.); 3Division of Cardiology, Department of Internal Medicine, Tongji Hospital, Tongji Medical College, Huazhong University of Science and Technology, Wuhan 430030, China; 15342267009@163.com

**Keywords:** prostate cancer, paeoniflorin, SRC, resistance, network pharmacology

## Abstract

**Background:** Despite advances in prostate cancer treatment, castration-resistant prostate cancer (CRPC) remains clinically challenging due to inherent therapy resistance and a lack of durable alternatives. Although traditional Chinese medicine offers untapped potential, the therapeutic role of paeoniflorin (Pae), a bioactive compound derived from Paeonia lactiflora, in prostate cancer has yet to be investigated. **Methods:** Using an integrative approach (network pharmacology, molecular docking, and experimental validation), we identified Pae key targets, constructed protein–protein interaction networks, and performed GO/KEGG pathway analyses. A Pae-target-based prognostic model was developed and validated. In vitro and in vivo assays assessed Pae effects on proliferation, migration, invasion, apoptosis, and tumor growth. **Results:** Pae exhibited potent anti-CRPC activity, inhibiting cell proliferation by 60% and impairing cell migration by 65% compared to controls. Mechanistically, Pae downregulated SRC proto-oncogene, non-receptor tyrosine kinase (SRC) mRNA expression by 68%. The Pae-target-based prognostic model stratified patients into high- and low-risk groups with distinct survival outcomes. Organoid and xenograft studies confirmed Pae-mediated tumor growth inhibition and SRC downregulation. **Conclusions:** Pae overcomes CRPC resistance by targeting SRC-mediated pathways, presenting a promising therapeutic strategy. Our findings underscore the utility of network pharmacology-guided drug discovery and advocate for further clinical exploration of Pae in precision oncology.

## 1. Introduction

Prostate cancer (PCa) is among the most frequently diagnosed malignancies in men worldwide, with incidence rates increasing in parallel with the aging population [1]. According to the latest cancer statistics, in the United States, PCa accounts for 29% of all male malignancies, ranking first in incidence and second in mortality [2]. Androgen deprivation therapy (ADT) remains an effective non-surgical treatment for localized PCa; however, the majority of patients eventually progress to the castration-resistant prostate cancer (CRPC) stage through various resistance mechanisms, including androgen receptor (AR) amplification, AR splice variants, intratumoral androgen synthesis, and neuroendocrine differentiation [3,4,5,6]. Currently, despite the availability of diverse therapeutic modalities for CRPC, the absence of standard treatment results in heterogeneous responses and poor long-term outcomes [7,8]. As our understanding of the intricate biology of CRPC deepens, encompassing AR signaling plasticity, tumor heterogeneity, epigenetic reprogramming, immune evasion mechanisms, and the dynamic interplay between tumor cells and the microenvironment, the development of more potent and targeted therapies becomes essential to improving treatment outcomes and enhancing the quality of life for CRPC patients [9,10].

In recent years, plant-derived compounds have gained recognition as promising therapeutic agents in cancer treatment [11,12]. These natural compounds, grounded in traditional medicinal practices, offer distinct advantages, such as reduced toxicity and the ability to target multiple pathways involved in tumor progression [13,14]. Compounds like paclitaxel [15] and camptothecin [16] have demonstrated potent anti-cancer effects and have become staples in clinical oncology. Research is increasingly focused on bioactive compounds with novel mechanisms of action, which hold significant potential to overcome drug resistance and improve patient outcomes [17]. This emerging field underscores the importance of integrating traditional knowledge with modern scientific methodologies to develop innovative plant-based therapeutic strategies for cancer management.

Paeoniflorin (Pae) is a primary bioactive component of Paeonia lactiflora [18,19]. Clinically, Pae has been used to treat inflammatory and autoimmune diseases [20], such as arthritis and inflammatory bowel disease, through the downregulation of pro-inflammatory cytokines and modulation of critical immune pathways [21,22]. Recently, Pae has attracted significant attention for its anti-cancer potential in various tumor types [23,24]. Mechanistic studies have demonstrated that Pae induces cell cycle arrest, promotes apoptosis, and modulates immune responses [25]. Pae also inhibits key tumor-promoting pathways, including PI3K/Akt and NF-κB signaling, and may enhance the efficacy of certain chemotherapeutic agents [22,26]. Despite these promising findings, the role of Pae in PCa, particularly CRPC, remains underexplored.

In this study, we explored the therapeutic potential of Pae in the treatment of CRPC. Using systematic target and pathway identification, we constructed a molecular interaction network and developed a prognostic model based on the potential target genes of Pae. Through a series of experiments, we demonstrated that Pae treatment in CRPC suppressed the expression of SRC, a gene strongly associated with PCa progression, by inhibiting its transcriptional activity. Notably, our findings confirmed that SRC inhibition significantly suppressed CRPC proliferation and metastasis in vivo, in organoid models, and in vitro. These results underscore the feasibility of Pae as a therapeutic agent for CRPC, paving the way for its integration into contemporary oncological treatment strategies.

## 2. Results

### 2.1. Identification of Paeoniflorin Potential Targets in Prostate Cancer

Paeoniflorin (Pae) is a natural compound derived from the roots of *Paeonia lactiflora*, with its chemical structure shown in Figure 1A. To investigate its potential targets in PCa, we integrated predicted target genes from the Swiss Target Prediction database with PCa-related genes identified from the GeneCards and OMIM databases. This analysis identified 10 potential target genes: *TP53*, *EGFR*, *AR*, *AKT1*, *ESR1*, *TGFBR2*, *SRC*, *NQO1*, *ALDH2*, and *ADH1B* (Figure 1B). Subsequently, we constructed a protein–protein interaction (PPI) network using the STRING database. The results revealed significant correlations among the target proteins, with the exception of *ALDH2* and *ADH1B*, which did not show notable interactions (Figure 1C).

To further investigate the potential mechanisms of Pae in PCa, we performed Gene Ontology (GO) analysis on the identified target genes. The top four pathways in the biological process (BP), cellular component (CC), and molecular function (MF) categories are presented in Figure 1D, with BP terms emphasizing prostate gland development. Additionally, KEGG pathway enrichment analysis identified 20 pathways, including “prostate cancer,” “pancreatic cancer,” and “bladder cancer,” as shown in Figure 1E.

In summary, we identified potential target genes of paeoniflorin in PCa and explored their associations with PCa progression, providing a foundation for further investigation into its therapeutic mechanisms.

### 2.2. Exploration of Clinical Characteristics of Ten Target Genes in Prostate Cancer

We next investigated the association of ten target genes with clinical characteristics in PCa patients. Utilizing the TCGA database, we initially compared the expression levels of these genes in PCa tissues versus adjacent non-cancerous tissues. This analysis demonstrated elevated *TP53* expression in tumor tissues relative to adjacent normal tissue, though mutational status was not assessed. Concurrently, significantly reduced expression of *ADH1B*, *ALDH2*, *EGFR*, and *TGFBR2* was observed (Figure 2A). Additionally, we examined the relationship between gene expression and clinical parameters, including T stage and Gleason score. Increased expression of *ADH1B* and *ESR1* correlated with higher T stages, whereas *ALDH2* demonstrated an inverse trend (Figure 2B). Moreover, patients with a Gleason score ≤7 exhibited higher expression levels of *ALDH2*, *NQO1*, and *TP53*. Conversely, *AR* expression levels increased alongside Gleason scores >7 (Figure 2C). Finally, we analyzed biochemical recurrence (BCR) survival in relation to the expression of these genes. Kaplan–Meier survival curves indicated that high expression levels of *AKT1*, *ALDH2*, *EGFR*, *ESR1*, *SRC*, and *TGFBR2* were associated with reduced survival rates over time. In contrast, elevated levels of *NQO1* and *TP53* were linked to a lower risk of biochemical recurrence (Figure 2D).

Collectively, these findings highlight the potential therapeutic value of Pae in PCa, as its target genes show a strong association with PCa progression and prognosis.

### 2.3. Establishment of a Prognosis Signature Based on Paeoniflorin Target Genes

To further investigate the relationship between 10 target genes and the prognosis of PCa, we applied univariate regression, LASSO analysis, and multivariate regression. Using the TCGA database, we constructed a prognostic signature based on the following risk score formula: 0.6361 × *SRC* − 0.3850 × *TP53* + 0.3696 × *ESR1.* While this signature demonstrated statistically significant predictive capability (AUC: 0.643, 0.720, and 0.638 for 1-, 3-, and 5-year survival, respectively), we recognize these values indicate room for improvement in discrimination accuracy (Figure 3A). Subsequently, we evaluated the model by stratifying patients into high- and low-risk groups according to the risk score. Survival analysis revealed a significantly higher risk of BCR in the high-risk group (Figure 3B). Furthermore, as risk scores increased, survival rates among PCa patients correspondingly declined (Figure 3C). The predictive capability of this signature was validated using the DKFZ PCa cohort (*n* = 118; accessed via cBioPortal, https://www.cbioportal.org/, accessed on 15 November 2024), yielding AUC values of 0.700, 0.659, and 0.659 for 1-, 3-, and 5-year predictions (Figure 3D). Moreover, multivariate Cox regression revealed that risk score, T stage, and Gleason score independently predicted BCR in PCa (Figure 3E). Based on these findings, we developed a comprehensive nomogram that incorporates these clinical parameters to predict survival outcomes and assist in clinical decision-making (Figure 3F).

Emerging evidence supports the feasibility of immunotherapy for PCa [27,28], emphasizing the necessity of understanding its interaction with the immune system. To examine the relationship between signature risk scores and immune infiltration, we employed the CIBERSORT algorithm, while acknowledging that such deconvolution approaches inherently assume that (1) cell-type signatures remain stable across biological conditions, and (2) bulk gene expression represents a linear combination of pure cell-type profiles. The results showed that high-risk groups exhibited greater infiltration of CD4^+^ T cells, naïve B cells, dendritic cells, and monocytes, whereas low-risk groups displayed higher activity of mast cells, NK cells, and helper T cells (Figure 3G). For clarity, immune cells were categorized into four groups: dendritic cells, lymphocytes, macrophages, and mast cells. Further analysis revealed that the signature was primarily associated with the infiltration of dendritic cells and mast cells (Figure 3H).

Gene mutations play a pivotal role in driving PCa progression, therapy resistance, and alterations in key signaling pathways [29]. We analyzed mutation differences between high- and low-risk groups, identifying the five most distinct mutations (Figure 3I). Notably, *TP53* mutations were significantly more frequent in the high-risk group, whereas mutations in *CASZ1*, *PTPRC*, *RNF213*, and *TTN* were more prevalent in the low-risk group.

In conclusion, we developed an effective prognostic model for PCa. This model is closely associated with prognosis, immune infiltration, and gene mutations in PCa patients, indicating the non-negligible role of Pae in PCa.

### 2.4. Molecular Docking of Paeoniflorin Target Genes

To investigate the therapeutic mechanism of Pae in PCa, molecular docking simulations were conducted using AutoDock 4.2.6 to model the binding interactions between Pae and its potential target proteins. Binding energy values from these simulations reflect the strength and stability of the interactions, with lower (more negative) values indicating stronger binding affinities [30]. The binding energy values were as follows: TP53 (−5.0 kcal/mol), AR (−2.68 kcal/mol), TGFBR2 (−6.58 kcal/mol), EGFR (−3.75 kcal/mol), AKT1 (−3.7 kcal/mol), ESR1 (−3.72 kcal/mol), SRC (−4.37 kcal/mol), and NQO1 (−5.0 kcal/mol) (Figure 4A–H). These findings demonstrate potential but relatively modest interactions between Pae and the target proteins, suggesting that Pae may exert therapeutic effects in PCa through modulation of the activity of these key genes.

### 2.5. Paeoniflorin Reduces SRC Expression Levels in Prostate Cancer

To further investigate the role of Pae in PCa, we initially employed the CCK-8 assay to assess its effect on the viability of 22Rv1 and C4-2 cell lines and to determine the optimal drug concentration for stimulation. The results indicated that Pae significantly inhibited the growth of both 22Rv1 and C4-2 cells in a dose-dependent manner, within the concentration range of 0 to 240 μM. The calculated IC_50_ values were 120.3 μM for 22Rv1 cells and 136.6 μM for C4-2 cells (Figure 5A,B).

We next examined the effect of Pae on the expression of ten previously identified target genes. Using the IC_50_ value, qPCR analysis revealed the most significant decrease in *SRC* mRNA levels following Pae treatment (Figure 5C,D). In accordance with these findings, Western blot analysis confirmed a substantial reduction in SRC protein expression after Pae treatment (Figure 5E). These findings suggest that SRC is the primary target of Pae in PCa.

To assess whether Pae influences the transcription of the *SRC* gene, we utilized a dual luciferase reporter system with the *SRC* gene promoter sequence inserted, which was sequence-verified and functionally validated. Our results demonstrated a marked reduction in firefly luciferase activity in the Pae-treated group compared to the DMSO control group. Notably, the extent of this inhibition was concentration-dependent, suggesting that Pae significantly suppresses *SRC* promoter transcriptional activity in PCa, resulting in the downregulation of *SRC* expression (Figure 5F). Data from a CRISPR-mediated genome-wide loss-of-function screen in PCa cell line revealed that *SRC* is a critical essential gene for cell survival (Figure 5G).

To investigate the relationship between *SRC* expression and PCa progression, we analyzed a cohort of 60 PCa patients from our hospital (Tongji PCa Cohort). mRNA levels of *SRC* were significantly higher in PCa tissues compared to adjacent non-tumor tissues (Figure 5H). Furthermore, elevated SRC expression correlated with more advanced tumor stages (Figure 5I). Notably, patients with high *SRC* expression had significantly lower biochemical recurrence-free survival rates than those with low *SRC* expression (Figure 5J).

Taken together, these results suggest that paeoniflorin exerts a potent inhibitory effect on *SRC*, an oncogene closely associated with PCa progression. These findings underscore the critical potential of paeoniflorin in the treatment of PCa.

### 2.6. Paeoniflorin Effectively Inhibits CRPC Cell Proliferation and Metastasis and Induces Apoptosis

To evaluate the therapeutic potential of Pae in CRPC, we investigated its effects on cell proliferation, metastasis, and apoptosis in CRPC cell lines through a series of functional assays. We assessed the impact of Pae on the proliferative capacity of CRPC cell lines (22Rv1 and C4-2) using CCK-8, EdU, and colony formation assays. Treatment with 120.3 μM and 136.6 μM Pae, respectively, followed by assessments of cell viability on days 1, 2, and 3, revealed a significant reduction in proliferative activity in both cell lines (Figure 6A). After 48 h of Pae treatment, the number of EdU-positive cells in the 22Rv1 and C4-2 cell lines decreased markedly (Figure 6B), and the colony formation rate was significantly lower in the Pae-treated group compared to the DMSO control group (Figure 6C). These results demonstrate that Pae effectively inhibits CRPC cell proliferation.

To investigate the effect of Pae on the metastatic phenotype of CRPC, we conducted Transwell and wound healing assays (Figure 6D,F). In the Transwell migration and invasion assays, 22Rv1 cells were treated with Pae (experimental group) or DMSO (control group). The results indicated that, compared to the control group, the number of cells migrating and invading through the membrane was significantly reduced in the experimental group, demonstrating that Pae effectively inhibits both migration and invasion of CRPC cells (Figure 6D). In the wound healing assay, images were captured at 0 h, 24 h, 48 h, and 72 h after creating the scratch. As shown in the figures, the cell migration rate increased over time; however, the migration rate of the Pae-treated group remained significantly lower than that of the control group (Figure 6E,F), suggesting that paeoniflorin effectively suppresses CRPC cell migration.

To assess whether Pae induces apoptosis in CRPC cells, we performed flow cytometric analysis using Annexin V-APC/PI double staining. A significantly higher proportion of apoptotic cells was observed in the Pae-treated group compared to the control group (Figure 6G,H). These findings suggest that Pae acts as a potent inducer of apoptosis in CRPC cells.

In summary, our in vitro experiments demonstrate that Pae effectively inhibits proliferation and metastasis while promoting apoptosis in CRPC cells. These results identify Pae as a promising therapeutic agent for treating CRPC.

### 2.7. Paeoniflorin Has Therapeutic Effects on Patient-Derived Organoids and Subcutaneous Xenograft Tumors in Nude Mice

Organoids have recently emerged as a novel model, providing a more accurate representation of drug effects compared to traditional in vitro cell culture models [31,32,33]. They better replicate the in vivo tumor characteristics and cell heterogeneity, providing a closer approximation of the tumor’s structural and functional properties while enabling the visualization of tumor growth [34]. In this study, we established PCa organoids from patient-derived tissue to evaluate the therapeutic effects of Pae. Light microscopy images revealed morphological changes in the organoids, with a tendency for rupture following Pae treatment (Figure 7A). Furthermore, the size of the organoids in the treatment group was significantly reduced compared to the control group (Figure 7B). Consistent with the in vitro results, the proliferative capacity of the organoids was markedly decreased following Pae treatment (Figure 7C), further supporting its inhibitory effect on PCa growth.

To further evaluate the therapeutic effects of Pae in CRPC, we established subcutaneous xenograft tumors by injecting two CRPC cell lines, 22Rv1 and C4-2, into the subcutis of nude mice. Once the tumor volume reached 50–80 mm^3^, treatment with Pae was initiated at a dose of 50 mg/kg, administered intraperitoneally every 3 days. Tumor growth was closely monitored and recorded throughout the treatment period. Compared to the control group, the Pae-treated group exhibited a significant reduction in tumor growth rate, and the final tumor volume was considerably smaller (Figure 7D,F). Upon completion of the experiment, the subcutaneous tumors were excised and weighed, revealing that the tumors from the Pae-treated group were significantly lighter than those in the control group (Figure 7E,G). These findings indicated that Pae effectively inhibited tumor growth in vivo.

IHC staining of subcutaneous xenograft tumors derived from two CRPC cell lines in nude mice revealed a significant reduction in SRC protein expression following Pae treatment (Figure 7H). Additionally, the observed decrease in Ki-67 expression in the treatment group indicated the inhibition of tumor proliferation (Figure 7I,J). H&E staining of tissues from the heart, liver, spleen, lungs, and kidneys of nude mice across all groups displayed normal histological features (Figure 7K), confirming the safety of Pae treatment in this model.

Collectively, these experiments demonstrated the therapeutic potential of Pae in both patient-derived organoids and subcutaneous xenograft tumors in nude mice. These findings highlight Pae as a promising candidate for the treatment of CRPC. Further clinical studies are required to validate its efficacy and safety in patients with CRPC.

## 3. Discussion

CRPC remains a significant clinical challenge due to its aggressive nature, limited treatment options, and resistance to conventional androgen deprivation therapies [35]. Despite advancements in targeted therapies, effective therapeutic targets are still lacking [36]. This study explores the therapeutic potential of Pae, a bioactive compound derived from TCM, for the treatment of CRPC. Through various analytical methods, we identified Pae’s target genes in PCa and demonstrated their association with disease progression and prognosis. Notably, our experiments confirmed that Pae inhibits PCa proliferation and metastasis by suppressing *SRC* transcription.

In recent years, there has been increasing recognition of the potential of TCM in cancer treatment [37,38]. Pae, a major bioactive component of *Paeonia lactiflora*, exhibits diverse pharmacological properties, including anti-inflammatory, immune-modulating, and antitumor effects [22,39]. For instance, Zhu et al. showed that Pae alleviates myotube atrophy induced by C26 colon cancer cell-conditioned medium [40], while Yang et al. demonstrated Pae’s ability to inhibit gastric cancer proliferation and metastasis via the RAS/MAPK signaling pathway [41]. Although the role of Pae in PCa has not been previously studied, these findings underscore its potential as a therapeutic candidate for conditions like CRPC.

Building upon prior research, our study clarifies the specific role of Pae in regulating critical pathways in CRPC. Mechanistic analyses revealed that Pae suppresses the expression of *SRC*, a non-receptor tyrosine kinase [42], by modulating its transcriptional activity. This downregulation significantly reduced CRPC proliferation and metastasis. Notably, Pae’s dual role in targeting both tumor growth and metastatic potential presents a novel strategy for utilizing natural compounds in cancer therapy [25,43,44,45].

A critical challenge in preclinical drug development is the selection of appropriate models. Traditional in vitro cell and in vivo mouse models often fail to accurately replicate human tumor responses [46]. Tumor-derived organoids have emerged as an advanced model, effectively mimicking the physiological and molecular characteristics of human tumors [47]. In PCa, organoids offer a transformative platform for studying tumor biology and drug responses, while preserving the tumor microenvironment and genomic heterogeneity [48]. By integrating network pharmacology with patient-derived organoid models [49], we evaluated Pae’s therapeutic efficacy, bridging the gap between preclinical and clinical applications. Brightfield microscopy revealed the rupture of 3D spherical structures in Pae-treated organoids, indicating antitumor effects. Organoid size, which correlates directly with cancer cell proliferation and survival, was significantly reduced following Pae treatment, suggesting Pae-induced cell cycle blockade. Luminescence assays further showed decreased metabolic activity, indicative of apoptosis or necrosis.

To further validate Pae’s antitumor effects in vivo, we established subcutaneous xenograft tumor models in nude mice. Continuous intraperitoneal injections of Pae resulted in a significant reduction in tumor growth. IHC analysis confirmed reduced expression of SRC and Ki-67 in Pae-treated tumors, while H&E staining showed no significant toxicity in major organs, demonstrating Pae’s favorable safety profile.

Despite the promising results, this study has several limitations. First, the precise mechanism by which Pae affects *SRC* requires further investigation, as our study primarily demonstrated its impact on transcription. Second, Pae’s role likely extends beyond the 10 predicted genes examined in our study, necessitating more extensive analyses. Finally, emerging models, such as patient-derived xenografts, could further strengthen the conclusions drawn in this study.

## 4. Materials and Methods

### 4.1. Chemical Compounds

Paeoniflorin was obtained from MedChemExpress (MCE, Monmouth Junction, NJ, USA), and its molecular structure was downloaded from PubChem (National Center for Biotechnology Information, Bethesda, MD, USA) [50].

### 4.2. Identification of Targets for Pae

Using the GeneCards (https://www.genecards.org/, accessed on 15 November 2024) and OMIM databases (https://omim.org/, accessed on 15 November 2024), 128 genes were identified as associated with PCa [51,52]. Potential drug targets for Pae were identified through Swiss Target Prediction (http://www.swisstargetprediction.ch/, accessed on 15 November 2024), yielding 100 Pae-related genes. A Venn diagram analysis revealed 10 shared targets between Pae and PCa.

### 4.3. GO and KEGG Enrichment Analysis

Gene Ontology (GO) [53] and Kyoto Encyclopedia of Genes and Genomes (KEGG) [54] pathway enrichment analyses were performed using the R package “enrichplot (v1.12.0).” The resulting enrichment data were visualized using the R package “ggplot2 (v3.3.5)”.

### 4.4. Establishment of Prognosis Signature

Biochemical recurrence (BCR) data for PCa patients were retrieved from the TCGA database via the UCSC Cancer Genomics Browser (https://genome-cancer.ucsc.edu/, accessed on 15 November 2024). A prognostic signature was developed using univariate regression, LASSO analysis, and [55,56]. The risk score was calculated using the following formula: Risk Score = Σi Coefficient × Expression. The predictive reliability of the Pae-related signature was assessed through time-dependent receiver operating characteristic (ROC) analysis. To validate the model’s performance, ROC analysis was conducted on an independent dataset from the Deutsches Krebsforschungszentrum (DKFZ). RNA-seq and clinical data for PCa samples from the DKFZ were obtained via cBioPortal (http://cbioportal.org, accessed on 15 November 2024) [57].

The proportions of 22 immune cell types were estimated using CIBERSORT, integrating these data into the gene expression matrix for analysis across different patient risk groups. Additionally, tumor mutation profiles between high- and low-risk groups were compared using the R package “maftools” [58].

### 4.5. Molecular Docking

The 3D structure of Pae was obtained from PubChem, and protein structures of target genes were sourced from the RCSB PDB database (https://www.rcsb.org/, accessed on 15 November 2024) [59]. Proteins ALDH2(pdb_00008dr9) and ADH1B(pdb_00001hsz) were excluded due to unavailable PDB structures to maintain docking accuracy. Docking simulations were conducted using AutoDock Vina, with AutoDock Tools for determining binding sites [60]. Docked complexes underwent 100 ns all-atom MD simulations. The grid box was centered on the binding site. Binding affinity was assessed based on docking energy, with lower energy values indicating stronger binding. The top docking results were visualized using PyMOL (v2.5.0) software.

### 4.6. Cell Culture

The cell lines used in this work are 22Rv1 and C4-2, originally purchased from ATCC (American Type Culture Collection, Manassas, VA, USA). Briefly, 22Rv1 and C4-2 cells were maintained in RPMI1640 medium. The cells were supplemented with 10% fetal bovine serum and 1% penicillin/streptomycin solution, and incubated in a humidified atmosphere containing 5% carbon dioxide at 37 °C.

### 4.7. Human PCa Tissues

The Tongji PCa cohort included 60 PCa patients from Tongji Hospital, and the tissues were obtained from the resected PCa tissue and the adjacent tissues of these patients during surgery. Fresh tissue samples were rapidly dissected into small fragments (typically 2–5 mm^3^) using sterile surgical blades on a chilled surface (4 °C) to minimize RNA degradation. The Clinical Trial Ethics Committee of Tongji Hospital granted ethical permission. We obtained written informed consent from all patients. (approval No. 2019CR101)

### 4.8. Extraction of Protein and Western Blot

Total protein was lysed on ice using RIPA lysis buffer (Solarbio, Beijing, China) containing protease and phosphatase blockers. Ultrasound was used to treat proteins, with a 3 s ultrasound program and 3 s intervals, for a total of 30 times. Then, centrifugation was performed at 13,000× *g* for 15 min at 4 °C, and the supernatant was retained as the protein sample. Protein concentration was measured using the BCA protein quantification kit (BOSTER, Wuhan, China). Then, protein loading buffer (Seven, Beijing, China) was added to the protein sample and boiled. The same amount of total protein in each sample was separated by SDS–polyacrylamide gel electrophoresis (SDS-PAGE) (PAGE Gel Fast Preparation Kit, purchased from EpiZyme, Shanghai, China), and then the protein in the gel was transferred to the PVDF membrane. The protein was sealed on the membrane with a rapid blocking solution, and then the membrane was placed in primary antibody (SRC Rabbit pAb and β-actin Rabbit mAb purchased from Abclonal, Wuhan, China) and incubated overnight at 4 °C. The next day, after washing the membrane with TBST, the membrane was incubated with HRP-conjugated secondary antibody (Abclonal, Wuhan, China) for 1 h. After washing again, the membrane was wetted with ECL chemiluminescence solution, and the target protein bands were visualized using a chemiluminescence imaging instrument (Bio Rad, Hercules, CA, USA). Semi-quantitative analysis and normalization on the results were performed using ImageJ (v1.53) software.

### 4.9. Luciferase Reporter Assay

The luciferase reporter plasmid containing wild-type or mutant SRC 3′ UTR was transfected into 22Rv1, C4-2 cells to obtain reporter zygotes. According to the supplier’s instructions, a dual luciferase reporter kit (Solarbio, D0011) was used to measure the relative luciferase activity of cells.

### 4.10. RNA Isolation and Real-Time Quantitative PCR

According to the operating protocol, total RNA was isolated using Total RNA Reagent (Trizol, ABclonal, Wuhan, China). Reverse transcription of total RNA was performed using Hifair III RT Buffer (Yeasen, Shanghai, China). Then, the Hieff qPCR SYBR Green Master Mix (Low Rox Plus) kit (Yeasen, Shanghai, China) was used to quantify mRNA levels. Relative gene expression was determined using the 2^−ΔΔCt^ method, with the housekeeping gene GAPDH as the normalized reference. Primer sequences are listed in Appendix A.

### 4.11. Cell Counting Kit-8 (CCK-8) Assay

The half-maximal inhibitory concentrations (IC_50_) of Pae on 22Rv1 and C4-2 cell lines, as well as the effect of Pae on cell activity at different intervention times, were determined using CCK-8 (MCE, USA). Untreated 22Rv1 and C4-2 cells (100 μL, 1 × 10^4^ cells/well) were inoculated onto a 96-well plate and incubated at 5% CO_2_ and 37 °C for 24 h. Afterwards, cells were treated with concentrations ranging from 0 to 240 nM and IC_50_ concentrations, while the control group was treated with 0.01% DMSO. After intervention at different times, 10 μL of CCK8 solution was added to each well of the plate and incubated for 2 h (5% CO_2_, 37 °C). The 96-well plate was gently stirred, and a microplate reader was then used (Model 680, BIO-RAD, USA) to measure the absorbance at 450 nm.

### 4.12. EdU Cell Proliferation Assay

Cells were inoculated into a 24-well plate (8 × 10^4^ cells/well) and cultured for 24 h and incubated at 5% CO_2_ and 37 °C for 24 h. Briefly, 22Rv1 and C4-2 cells were treated with Pae at their respective IC50 concentrations (120.3 μM for 22Rv1 and 136.6 μM for C4-2) prior to assessment using the EdU Cell Proliferation Assay Kit (Beyotime Biotechnology, Shanghai, China), which was used to determine the proliferating cells, and photos under blue light were captured using a fluorescence microscope (Bio-Rad, CA, USA) to identify Hoechst cells, while photos under red light were captured to obtain EdU cell photos. The percentage of proliferating cells was calculated by dividing the number of Edu cells by the number of Hoechst cells.

### 4.13. Colony Formation Assay

Briefly, 22Rv1 and C4-2 cells (treated with Pae 120.3 μM and 136.6 μM, respectively, and untreated controls) were seeded at a density of 800 cells/well in 6-well plates (Corning). After incubating for 10 days (medium replacement every 3 days), colonies of ≥50 cells were counted and compared with the clone numbers of the two groups of cells.

### 4.14. Transwell Migration and Invasion Assay

The Transwell cell migration and invasion assay was used to analyze the migration and invasion ability of 22Rv1 cells through Corning Transwell (8.0 mm polycarbonate membrane). For invasion assays, inserts were coated with Matrigel. Then, 22Rv1 cells (1 × 10^5^ cells) were seeded into the upper chamber of Transwell inserts and treated with 120.3 μM Pae for 48 h in a 24-well culture dish containing FBS-free medium in the upper chamber and 10% FBS medium in the lower chamber. Then, the cells were fixed with 4% paraformaldehyde and stained with crystal violet (Beyotime), and the migrated and invaded cells were observed under a computerized microscope.

### 4.15. Wound Healing Assay

First, the inserts were placed on 6-well plates, and the processed cells were inoculated into the insert in the middle of the dish. After the cells had grown to the full insert area, the insert was removed with tweezers to create scratches. After washing three times with PBS to remove debris, cells were treated with Pae at concentrations of 120.3 μM (22Rv1) or 136.6 μM (C4-2) in serum-free RPMI-1640 medium for 72 h. The changes in the recovery of injured areas by migrating cells were recorded under an optical microscope (10×) (Olympus, Tokyo, Japan). Microphotographs were taken every 24 h, and the wound healing area was assessed using ImageJ software.

### 4.16. Apoptosis Flow Cytometry Assay

According to the manufacturer’s instructions, apoptotic cells were directly measured using an Annexin V-APC/PI dual staining cell apoptosis assay kit (KGA, Nanjing, China) and analyzed by flow cytometry. FlowJo software (v10.8.1) was used to calculate the percentage of early and late apoptotic cells, as well as necrotic or live cells.

### 4.17. Organoid Models

The PCa tissue of the patient was dissected, chopped, and dissociated using specialized culture medium and then treated with TrypLE (Gibco, Billings, MT, USA) and Y-27632. Luminal cells were treated by FACS sorting (Aria II, BD) and centrifuged before plating in Matrigel. An organoid formation assay was initiated with 2000 cells per well, and organoid characteristics were evaluated on day 14.

For the organoid viability assay, organoids were cultured in 96-well plates and exposed to drugs as specified. Fresh medium was replaced every other day, and ATP levels were measured using CellTiter-Glo 3D Reagent (Promega, Madison, WI, USA).

### 4.18. Experimental Animals

Male BALB/c nude mice aged 8 weeks were purchased from Hunan SJA Laboratory Animal Co., Ltd. Mice were kept at the SPF experimental animal center and allowed to adapt to the environment for one week. To observe the effect of Pae on tumor growth in vivo, nude mice were randomly divided into two groups (*n* = 5). Each mouse was subcutaneously injected with 3 × 10^6^ 22RV1 cells (100 μL PBS solution) into the axilla. After the subcutaneous transplant tumor grew to a volume of 50–80 mm^3^, mice were treated with intraperitoneal injection of Pae (50 mg/kg) or DMSO (control). Treatment was performed once every 3 days, and the volume of the transplanted tumor during this period was observed and recorded. The tumor size was calculated using the following formula: Volume = 1/2 × (Length × Width^2^). After 13 days of treatment, the subcutaneous tumor tissues of all groups were dissected, photos were taken, and they were then weighed. Mouse tumor samples were collected for immunohistochemistry (IHC) and hematoxylin–eosin (H&E) staining. All animal operations were approved and supervised by the Ethics Committee of Tongji Hospital Experimental Animal Center.

### 4.19. IHC and H&E Analysis

To evaluate the effect of Pae on tumor protein expression and the safety on the liver, spleen, lungs, and kidneys in nude mice, the tumor or the liver, spleen, lungs, and kidneys were embedded in paraffin and sectioned at 5 μm. For IHC, tissue sections were incubated with corresponding antibodies, followed by incubation with HRP enzyme-labeled secondary antibodies. Finally, DAB was used for color visualization, and images were observed and captured under a fluorescence microscope. For the H&E assay, tissue sections are dewaxed, rehydrated, stained with hematoxylin–eosin, and observed and captured under a fluorescence microscope.

### 4.20. Survival Prognosis Analysis

To evaluate the prognostic significance of target genes, we performed survival analysis using clinical data from the TCGA-PRAD cohort. The BCR survival data were retrieved from the UCSC Xena browser (https://xenabrowser.net, accessed on 15 November 2024). Tumor patients were stratified into high- and low-expression groups based on median expression levels of the target gene. Kaplan–Meier survival curves were generated using the R package “survival”, with statistical significance assessed by the log-rank test. Univariate Cox proportional hazard regression was employed to calculate HR with 95% confidence intervals. All analyses were conducted in R v4.2.3.

### 4.21. Statistical Analyses

The mean ± SD represents the data results of at least three independent experiments. All statistical analyses were conducted using GraphPad Prism (V.8.0.3, San Diego, CA, USA). Kaplan–Meier survival analysis was used to evaluate the association between gene expression and the probability of survival in multiple tumors. *P*-value, hazard ratio (HR), and 95% confidence interval (CI) were determined by the log-rank test. Statistical significance was assessed using bilateral unpaired Student’s *t*-test or one-way ANOVA, followed by Dunnett’s post hoc test. Spearman correlation analysis was used to detect gene co-expression. For functional experiments, the legend in the relevant figure represents the specific statistical tests used. *p*-value < 0.05 was considered statistically significant.

## 5. Conclusions

In conclusion, this study highlights Pae’s therapeutic potential for CRPC treatment. Through network pharmacology and enrichment analyses, we identified *SRC* as a critical target of Pae. Pae modulates both the transcription and protein levels of *SRC* to inhibit CRPC progression. Comprehensive in vitro and in vivo experiments demonstrated Pae’s efficacy in inhibiting cell proliferation, migration, and invasion while promoting apoptosis in CRPC cells. Patient-derived organoid and xenograft models further validated Pae’s antitumor activity and safety. These findings establish Pae as a promising candidate for CRPC therapy, thereby paving the way for future clinical applications.

## Figures and Tables

**Figure 1 pharmaceuticals-18-01241-f001:**
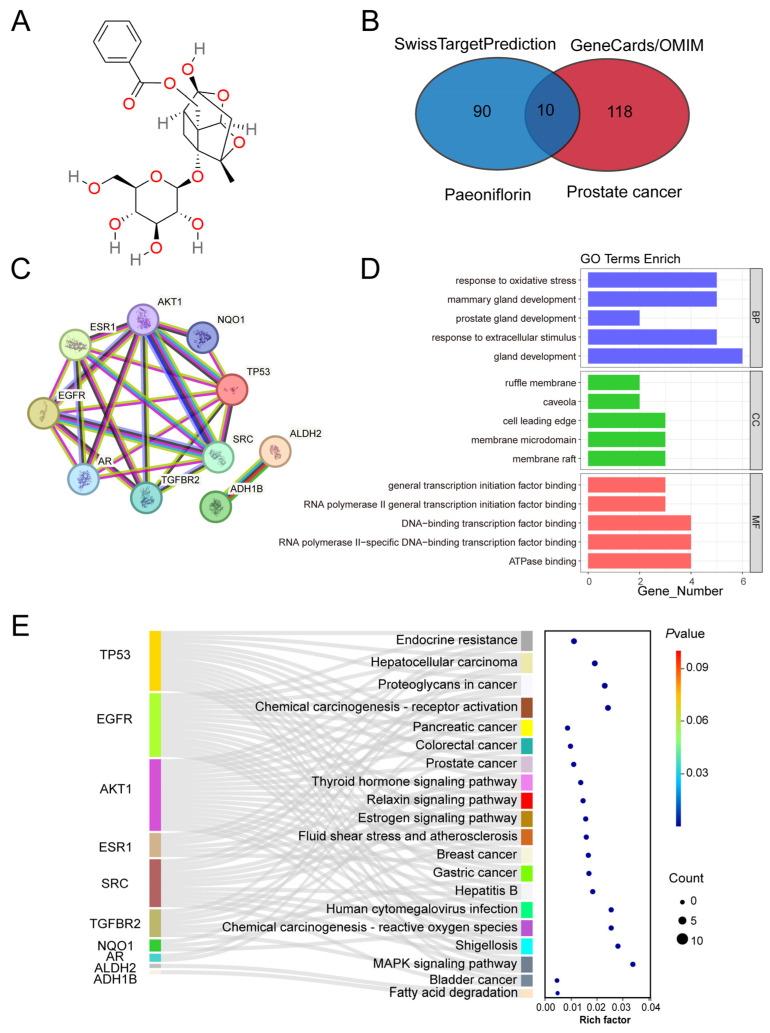
Identification of paeoniflorin potential targets in prostate cancer: (**A**) Pae’s chemical structure; (**B**) Venn diagram displaying the intersection targets of Pae and prostate cancer; (**C**) PPI network of Pae and prostate cancer intersection targets (nodes represent proteins, and edge represents protein–protein association); (**D**) GO enrichment analysis showing top BP, CC, and MF functional attributes of Pae’s targets against prostate cancer; (**E**) Sankey diagram for KEGG enrichment analysis of top 20 signaling pathways of Pae against prostate cancer.

**Figure 2 pharmaceuticals-18-01241-f002:**
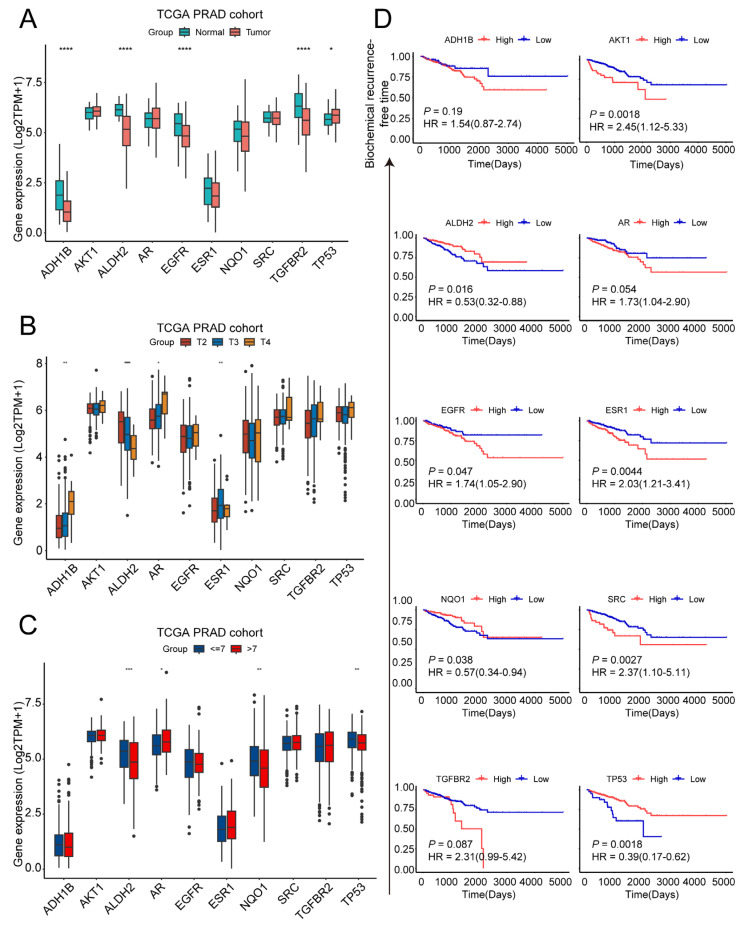
Exploration of clinical characteristics of ten target genes in prostate cancer: (**A**) the expression levels of 10 target genes in prostate cancer tissues and adjacent normal tissues in the TCGA PRAD cohort; (**B**) the expression levels of 10 target genes in prostate cancer tissues of different T stages (T2, T3, and T4 stage) in the TCGA PRAD cohort; (**C**) the expression levels of 10 target genes in prostate cancer tissues of different Gleason scores (Gleason score ≤ 7 and Gleason scores > 7) in the TCGA PRAD cohort; (**D**) survival analysis predicting the relationship between gene expression patterns of 10 target genes and patient biochemical recurrence-free time in PRAD. * *p* < 0.05, ** *p* < 0.01, *** *p* < 0.001, **** *p* < 0.0001.

**Figure 3 pharmaceuticals-18-01241-f003:**
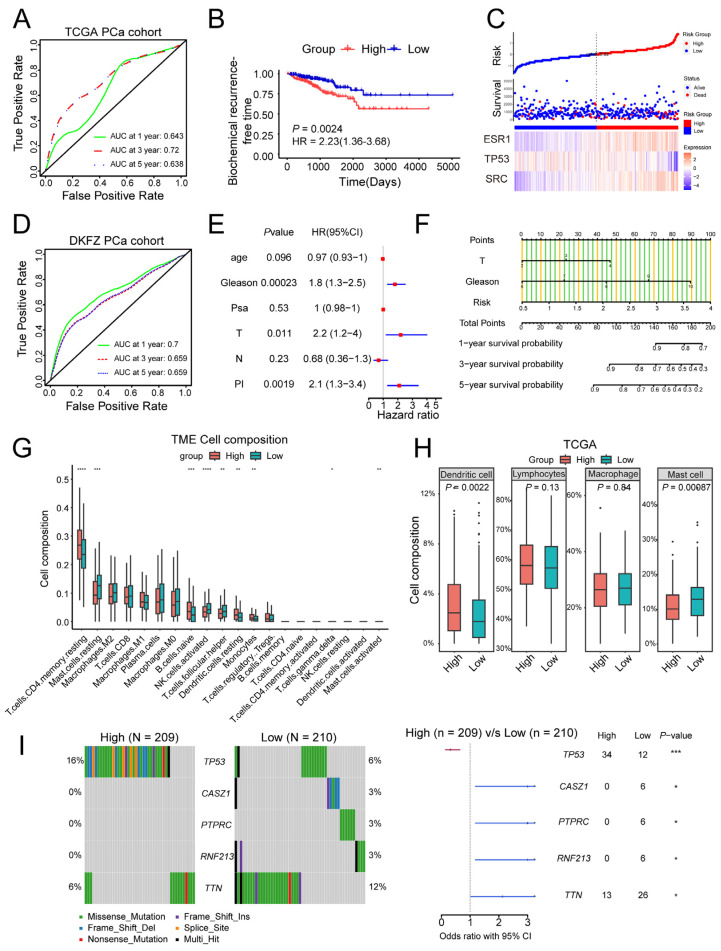
Establishment of a prognosis signature based on Paeoniflorin target genes: (**A**) receiver operating characteristic (ROC) curve constructed using the TCGA PCa cohort; (**B**) biochemical recurrence-free survival curves of patients with high- and low-risk in the TCGA PCa cohort; (**C**) the survival status distribution was analyzed based on the Pae-related signature; (**D**) receiver operating characteristic (ROC) curve constructed using the DKFZ prostate cancer cohort; (**E**) multivariate Cox regression models were used to analyze the associations between risk score, clinical factors, and PCa prognosis; (**F**) nomogram for patients predicting survival outcomes. T stage, Gleason score, and risk are marked as “points.” The total points by adding the three points can predict survival outcomes; (**G**) the CIBERSORT algorithm was utilized to evaluate and compare the infiltration of immune cells between the high- and low-risk score groups; (**H**) immune cells were categorized into four main types: dendritic cells, lymphocytes, macrophages, and mast cells. We then compared the differences in infiltration levels between the high- and low-risk score groups; (**I**) the differences in gene mutations between the high- and low-risk score groups were compared using waterfall plots. * *p* < 0.05, ** *p* < 0.01, *** *p* < 0.001, **** *p* < 0.0001.

**Figure 4 pharmaceuticals-18-01241-f004:**
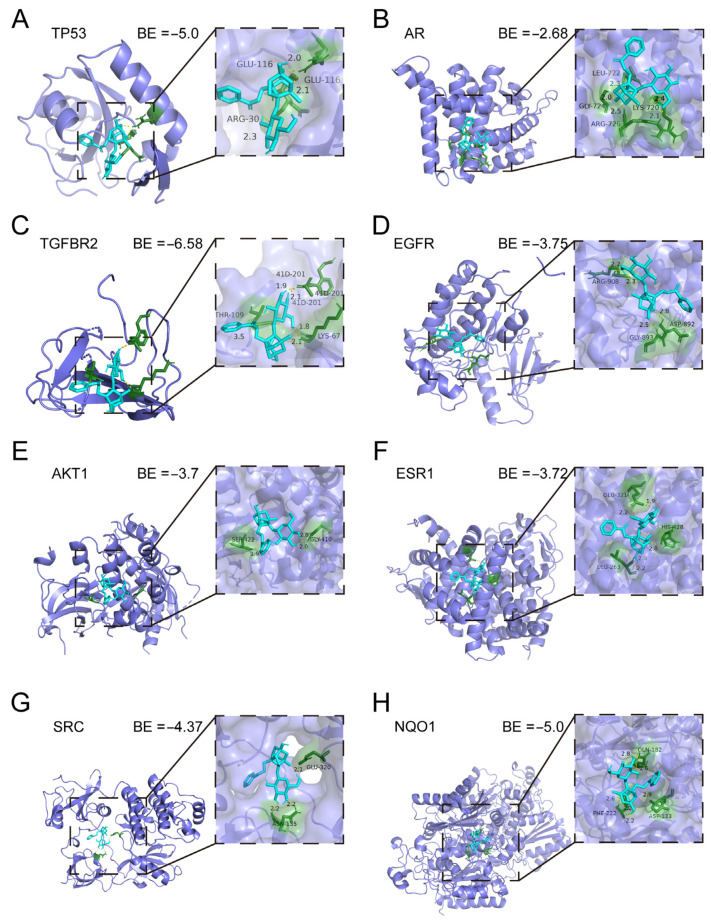
Molecular docking of Paeoniflorin target genes: Three-dimensional and two-dimensional docking patterns and interactions of (**A**) paeoniflorin–TP53; (**B**) paeoniflorin–AR; (**C**) paeoniflorin–TGFBR2; (**D**) paeoniflorin–EGFR; (**E**) paeoniflorin–AKT1; (**F**) paeoniflorin–ESR1; (**G**) paeoniflorin–SRC; (**H**) paeoniflorin–NQO1. Purple: protein encoded by the target gene, Blue: paeoniflorin, Green: binding sites.

**Figure 5 pharmaceuticals-18-01241-f005:**
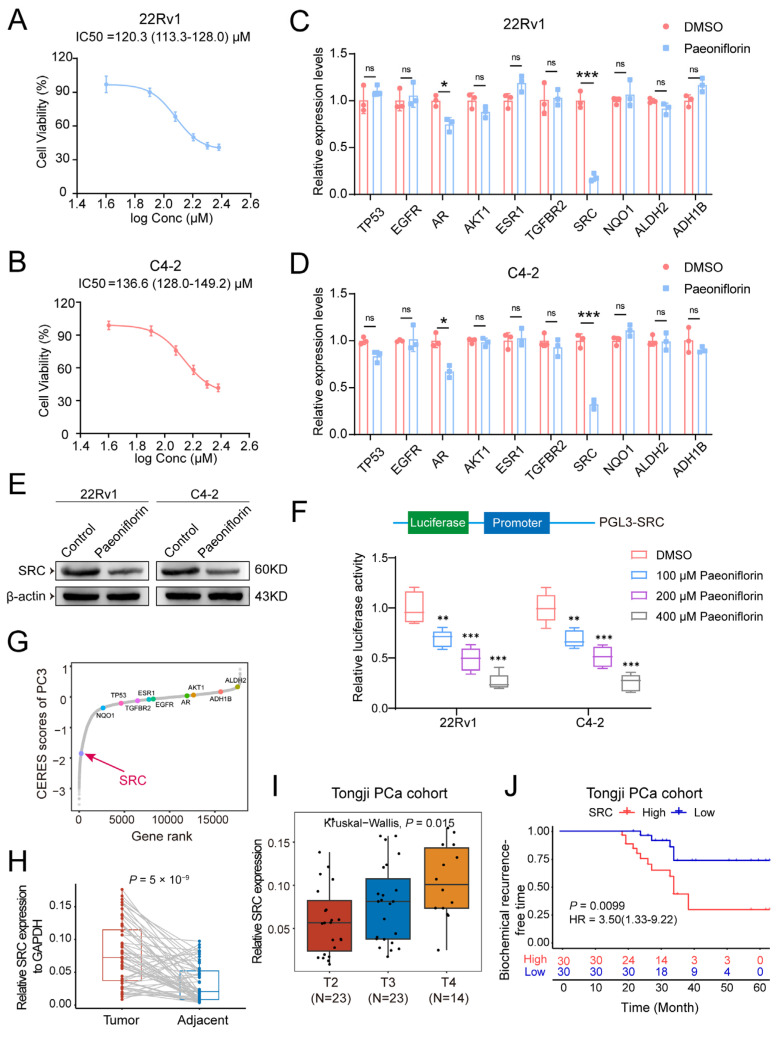
Paeoniflorin reduces SRC expression levels in PCa cell lines and PCa cohort tissues: (**A**,**B**) different concentrations of Pae were incubated with 22Rv1 and C4-2 cells for 48 h, and the effect of Pae on the cell viability was assessed by CCK-8 assay; (**C**,**D**) relative expression of mRNA levels of 10 target genes in 22Rv1 and C4-2 cells treated with Pae; (**E**) WB detection of relative expression of SRC protein levels in 22Rv1 and C4-2 cells treated with Pae; (**F**) quantification of luciferase activity in 22Rv1 and C4-2 cells with different concentration Pae treatment; (**G**) genome-wide loss-of-function screen in PC-3 identified essential genes including SRC and AR for cell survival. Lower scores indicate higher dependency on the gene for cell viability. SRC is highly expressed in primary prostate tumors; (**H**) relative expression of mRNA levels of SRC in tumor tissue and adjacent tissues of patients in Tongji PCa cohort; (**I**) relative expression of mRNA levels of SRC in PCa patients with different T stages in Tongji PCa cohort; (**J**) biochemical recurrence-free survival curves of patients with high and low SRC expression in Tongji PCa cohort. Statistical analysis: two-tailed Student’s *t*-tests were employed for statistical significance, with significance levels indicated as * *p* < 0.05, ** *p* < 0.01, *** *p* < 0.001, ns: not significan.

**Figure 6 pharmaceuticals-18-01241-f006:**
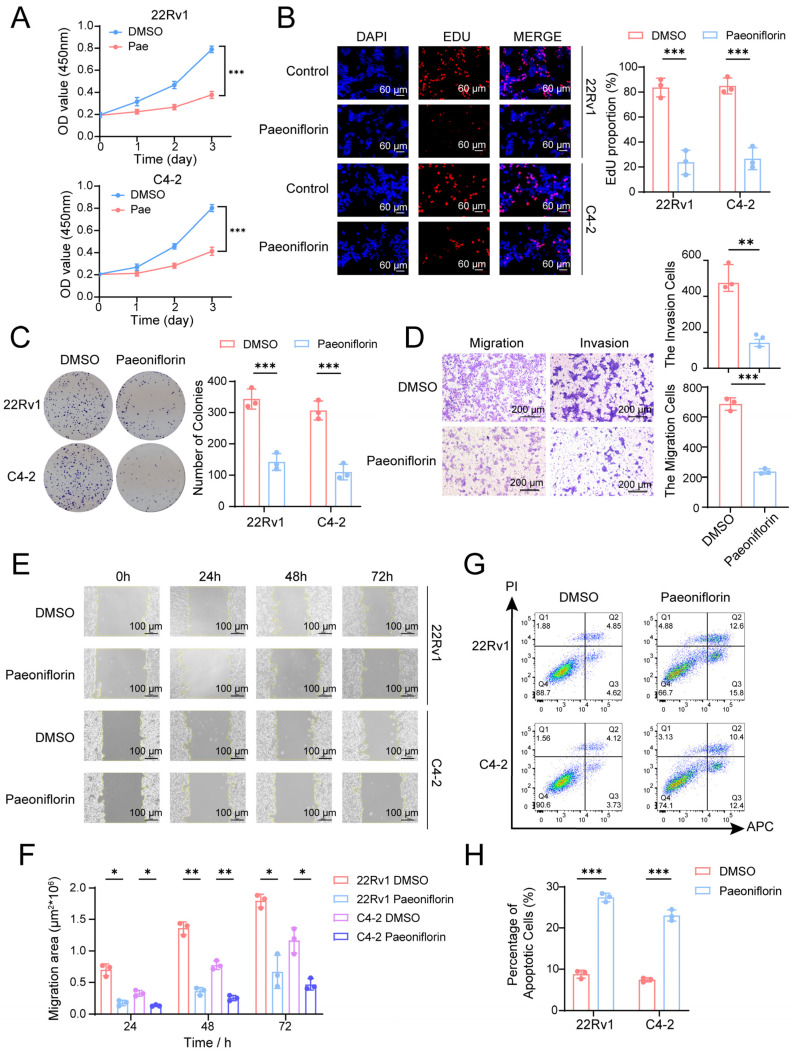
Paeoniflorin effectively inhibits PCa cell proliferation and metastasis and induces apoptosis: (**A**) 22Rv1 and C4-2 cells were treated with Pae for 1, 2 or 3 days, followed by CCK8 assay to evaluate cell viability; (**B**) 22Rv1 and C4-2 cells were treated with Pae for 2 days, followed by EdU experiments to evaluate cell proliferation; (**C**) the impact of Pae on the colony formation of 22Rv1 and C4-2 cells; (**D**) changes in migratory and invasive capabilities in 22Rv1 cells treated with Pae; (**E**,**F**) changes in wound healing ability of 22Rv1 and C4-2 cells treated with Pae on days 0–3; (**G**,**H**) the apoptosis rate of 22Rv1 and C4-2 cells after Pae intervention for 48 h were measured by flow cytometry. Statistical analysis: All data points were evaluated for statistical significance using two-tailed Student’s *t*-tests. Compared to DMSO (0.01%), * *p* < 0.05, ** *p* < 0.01, *** *p* < 0.001.

**Figure 7 pharmaceuticals-18-01241-f007:**
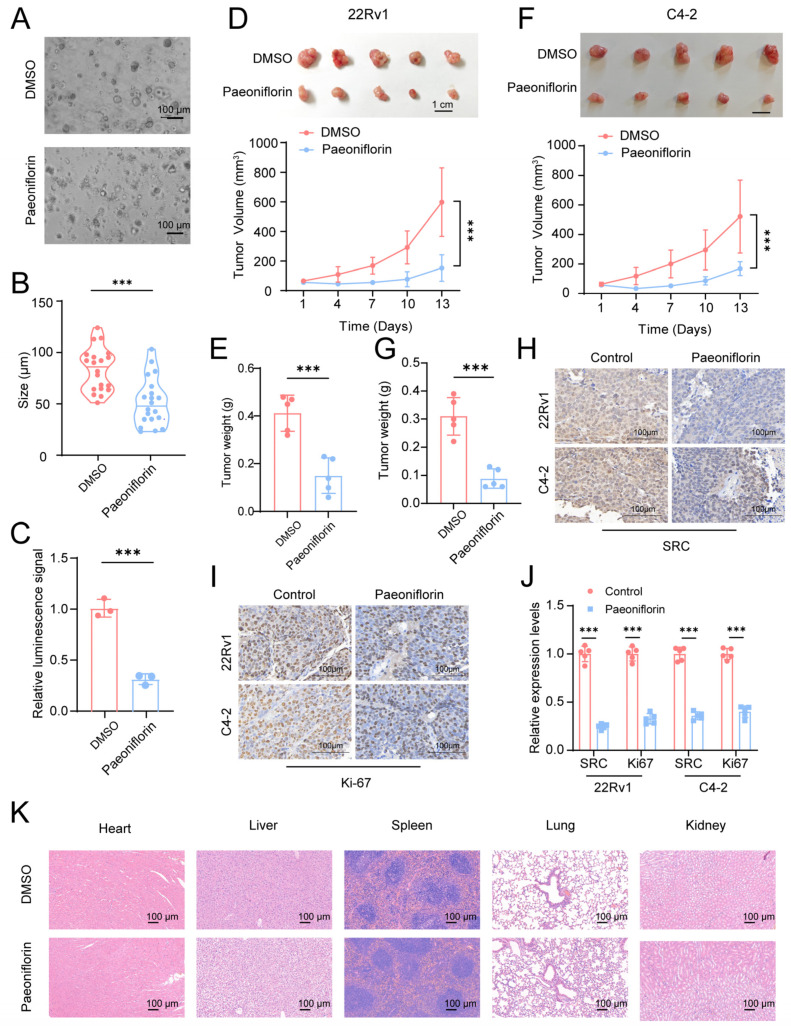
Paeoniflorin has therapeutic effects on patient-derived organoids and subcutaneous xenograft tumors in nude mice: (**A**) light microscopy images revealed morphological changes in the organoids; (**B**) changes in organoid size of Pae-treated group (*n* = 20); (**C**) changes in luminescence signals indicating metabolic activity and cell survival rate (*n* = 3); (**D**,**F**) top: images of excised 22Rv1 (**D**) and C4-2 (**F**) xenograft tumors in nude mice treated with DMSO or Pae, bottom: tumor volume growth curve of 22Rv1 (**D**) and C4-2 (**F**) xenograft tumor in nude mice treated with DMSO or Pae (*n* = 5); (**E**,**G**) quantification of tumor weight in excised 22Rv1 (**E**) and C4-2 (**G**) xenograft tumors (*n* = 5); (**H**,**I**) IHC staining for SRC (**H**) and Ki67 (**I**) in 22Rv1 and C4-2 xenograft tumors; (**J**) quantitative analysis of SRC and Ki67 expression; (**K**) H&E staining for evaluation of the safety of DMSO or Pae on the heart, liver, spleen, lung and kidney. Statistical analysis: All data points were evaluated for statistical significance using two-tailed Student’s *t*-tests. Compared to DMSO, *** *p* < 0.001.

## Data Availability

The data presented in this study are available on request from the corresponding author. The data are not publicly available due to being part of an ongoing study.

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
