# Peer review of "Network Pharmacology and Experimental Validation Identify Paeoniflorin as a Novel SRC-Targeted Therapy for Castration-Resistant Prostate Cancer"

_pharmaceuticals, 2025, doi:10.3390/ph18081241_

Round 1
Reviewer 1 Report
Comments and Suggestions for Authors
The author has novel study in “Network Pharmacology and Experimental Validation Identify 2 Paeoniflorin as a Novel SRC-Targeted Therapy for Castration-3 Resistant Prostate Cancer”. The author has justified their study in multiple experimental approaches along with in-depth supportive literature. This study could be beneficial for many researchers. This paper can be accepted after the major revision. However, further discussion on a practical application basis could enhance the strength of your paper:
Abstract
- Include a numerical value, such as (folds or % increase compared to control) instead of a general statement only, which will increase the impact of your abstract.
- Add more keywords, such as resistance, your main test, and main mechanism of action
- Define the abbreviation SRC in the abstract
Introduction
- Line 43, add some mechanisms that are responsible for resistance.
- Line 44, the statement is controversial, since multiple chemotherapies, immunotherapies, hormonal therapy or surgery could be used for CRPC treatments. Please clarify it, as standard treatment is not specified.
- Line 46, Please add more detail on complex biology.
- Please provide the abbreviation in beginning and use it throughout the manuscript. Example: PCa and CRPC in Line 38 and Line 71. Please unify all over your manuscript
Methods
- Please add a reference to GeneCards, OMIM databases, and Swiss Target Prediction.
- Please add a reference to univariate regression, LASSO analysis, multivariate regression, and R package "maftools."
- Provide PBD reference for ALDH2 and ADH1B.
- What were the charges and grid that you selected for the autodock? Please insert it in section 2.5.
- Line 126, how to discretize the tissue into single cell as well as incorporate cell culture conditions
- Please make more details in the method of sections 2.11, 2.12, 2.13, .14 and 2.15 (such as volume of treatments, drug concentration of treatments.
- Details of invasion assays should clarify whether Matrigel or equivalent barriers were used to distinguish invasion from migration.
Results and discussion
- The author has mentioned DMSO in the results, but it is not clear in the method section. Please clarify its % in cell migration, apoptosis, and other experiments.
- Incorporate the solubility of Paeoniflorin in water and 0.01% DMSO.
- Section 3.1 (Target Identification): While the integration of database predictions is reasonable, the authors should discuss the risk of false positives inherent in in silico target prediction and justify their selection criteria more clearly.
- Section 3.3 (Prognostic Signature): The AUC values reported are modest (0.643–0.720). This limitation should be acknowledged explicitly. Details of the validation cohort (sample size, clinical characteristics) are also insufficiently described. Please elaborate it.
- Validation in DKFZ is well performed, but sample sizes and cohort details are missing. Add these.
- The CIBERSORT-based immune infiltration analysis is interesting but assumes bulk RNA-seq deconvolution is valid—this assumption should be discussed.
- Docking energies are only modestly negative (e.g., –2.68 to –6.58 kcal/mol), suggesting relatively weak binding. The text should caution readers that these are predictive, not definitive interactions.
- The dual luciferase assay shows Pae inhibits SRC promoter activity but lacks controls for off-target effects or specificity. Suggest adding or discussing this.
- The CRISPR screen evidence (Fig. 5G) is correlative—SRC dependency is known but does not directly prove Pae acts only via SRC. Please clarify it.
- Organoid sample sizes (n) are not consistent across text and figures—standardize and clarify.
- Incorporate the scale bar to Figure 7J.
- The citation of the figure in the manuscript, “some are italic, some are normal” please unify it.
- Figure 6E, scratch wound assay or migration assay, represented a similar number of cells (cell compactness) in the Pae-treated group at 0 and 72 h. These results could also be related to cell cytotoxicity, which makes your results controversial. Could you please clarify it.
Reviewer 2 Report
Comments and Suggestions for Authors
- Researchers used in silico, in vitro, ex vivo and animal models to study the potential targets for Castration-Resistant Prostate Cancer and evaluated the anticancer activity of Paeoniflorin on Castration-Resistant Prostate Cancer. Identified SRC as a possible target for Paeoniflorin.
- The study was well designed. Results were presented and discussed very well. Conclusions were made based on experimental results.
- Improve the font size in figures: example in Fig 6G, the font size in the flow cytometry graphs are very small. Increase the size so that the % population in each quadrant are visible.
- Check the results section thoughly to describe the experiments used. example: in the section 3.7,
"Consistent with the in vitro results, the proliferative capacity of the organoids was markedly decreased following Pae treatment " indicate which assay was used to measure the proliferation.
- For figures 7H and 7I, provide quantitative analysis from the images.
Reviewer 3 Report
Comments and Suggestions for Authors
In this manuscript, Xu et al. interrogate the effects of Pae on CRPC using a diverse range of in silico and in vitro methodologies. Overall, this was an excellent manuscript that provided evidence that supports the claims of the paper. The in vitro studies were particularly good.
However, I have one concern: How exactly were the survival analysis graphs generated? I am focused here on Figures 2D and 3B. The Methods did not mention survival analysis. What was the source of this data? Did the authors actually perform these studies? Please provide a robust explanation.
Reviewer 4 Report
Comments and Suggestions for Authors
Plant-derived anti-cancer compounds are numerous, and, obviously, many are to be discovered in the future, thus the beautiful flowers like peonies should be also thoroughly investigated. In line with this, the manuscript "Network Pharmacology and Experimental Validation Identify Paeoniflorin as a Novel SRC-Targeted Therapy for Castration-Resistant Prostate Cancer" presents a number of potentially interesting findings (like downregulation of SRC), which may be useful anti-cancer drug design, therefore, it is possible to accept the manuscript for publication with the provision of a number of significant improvements. First, it should be stated in the introduction that the oncological potency of Paeonia root extracts were suspected long ago and in fact these were used to treat oncology patients, although without proper clinical trials. In contrast, it is not correct to state that "paeoniflorin (Pae), a primary bioactive component of Paeonia lactiflora, has long been recognized in traditional Chinese medicine (TCM) for its therapeutic properties" since traditional medicines deal with mixtures of natural bioactive ingredients, not pure substances.
Other points to consider:
In methods, the descriptions like "place the membrane in primary antibody" look like manuals, this is not traditional for scientific papers describing novel results, please change into "the membrane was placed" etc throughout the manuscript.
There are some typos like "computeried microscope", "t the pre". Please correct accordingly.
Statements like "TP53 was highly expressed in cancerous tissues" are misleading since cancer cells differ greatly in TP53 status, they may be positive, negative or contain oncogenic mutations in the gene.
The portion "Collectively, these findings confirm that the expression of these target genes is strongly linked to PCa progression and prognosis, highlighting the potential therapeutic value of Pae in Pca" places the horse behind the wagon, it should be rewritten in a more logical order.
"For clarity, immune cells were categorized into four groups: dendritic cells, 337
lymphocytes, macrophages, and mast cells" - this classification is too primitive and also misleading, fore example, macrophages may be classified into several populations with either pro- or anticancer properties.
The TTN gene is strictly muscle-specific, it is unclear how it can affect anything concerning PCA progression.
The statement "prognostic model for PCa based on ten Pae target gene" is not justified, maximally this may be suitable for predicting Pae's potential for PCA sensitivity, would Pae be ever approved for medical use.
Docking results are naive: "favorable binding interactions between Pae and these target proteins, suggesting that Pae may exert therapeutic effects in PCa through modulation of the activity of these key genes". In fact, these attempts do not present any serious mechanistic keys.
Fig. 1. Pae' s structural formula is of poor quality, please look how beautifully other people draw this interesting substance.
"fig 5 a, b – narrow range" unclear, please describe in more detail.
"proliferation and metastasis in figs 6d, 6e" and elsewhere. In fact, these results do not show any true effects in a metastasis model (in vivo).
The dose "500 mg/kg" is huge. Is this correct?
Interactions map from STRING is useless, all members are interconnected (in facts, the interactome is much more complex).
Round 2
Reviewer 1 Report
Comments and Suggestions for Authors
The manuscript has been drastically improved. Overall, this manuscript can be accepted after the minor revision. I would request the author to rewrite the abstract. (Emphasize the drawback or loophole in 1-2 lines, mention the method in short, then mainly focus on results, like mRNA expression, tumor suppression, or other results, and finally include your suggestion as well as it can be as an alternative approach for treatments.
Reviewer 4 Report
Comments and Suggestions for Authors
The response to criticism on aesthetic features of the structural formula of Pae is difficult to accept as satisfactory. That is approximately one hour of work even for an unexperienced user of chemical graphics software. Other significant points appear to be fixed. Therefore, it would be logical to recommend the manuscript for publication after redrawing Fig. 1A.
